# Fatigue Performance Analysis of Welded T-Joints in Orthotropic Steel Bridge Decks with Ultrasonic Impact Treatment

**DOI:** 10.3390/ma16186196

**Published:** 2023-09-13

**Authors:** Yizhou Liu, Wenhua Huang, Banhai Yu, Zhihao Chen, Ping Wang

**Affiliations:** 1School of Ocean Engineering, Harbin Institute of Technology at Weihai, Weihai 264209, China; 15519220415@163.com; 2Wuhan 2nd Ship Design and Research Institute, Wuhan 430200, China; 3School of Civil Engineering, Harbin Institute of Technology, Harbin 150001, China

**Keywords:** ultrasonic impact treatment (UIT), T-joints, grain refinement, traction structural stress, fatigue performance, stress concentration

## Abstract

This study aims to assess the effect of ultrasonic impact treatment (UIT) on the exterior weld seam of S355J2 T-joints used in orthotropic steel bridge decks. The microstructure and mechanical behavior of T-joints after UIT was investigated in this study. Fatigue tests of T-joints before and after UIT were performed. The stress concentration at the interior and exterior weld toe of T-joints was considered using the traction structural stress method. The results showed that hardness increases by 10% due to the localized grain refinement caused by UIT. UIT significantly improves the fatigue life of T-joint specimens by 350% and 150% at stress ratios of 0.1 and 0.3, respectively. As the transition angle between the weld profile and the base metal profile increases, the stress concentration factor decreases.

## 1. Introduction

Orthotropic steel bridge decks have been widely used worldwide due to their light weight, high bearing capacity, and fast construction speed. The fatigue crack problem of the bridge deck is more prominent due to factors such as its structural system, stress characteristics, processing and manufacturing, and service environment. The most vulnerable area for fatigue cracking is the welded joint, where different profiles are connected through welding. This greatly impacts the operational quality, durability, and safety of the structures [1]. Therefore, it is very necessary to optimize the treatment of the welded joints for the fatigue cracking problem of the bridge decks. Currently, the main methods for optimizing the welded joints treatment include aging treatment, heat treatment, explosive treatment, shot peening, and ultrasonic impact treatment (UIT). Among these methods, UIT is a more ideal post-weld treatment method because it has the advantages of lightweight, flexibility, high efficiency, and low cost. It has been widely applied in the regulation of residual stresses in welded joints and in the improvement of the fatigue life of bridge decks [2,3]. UIT strengthens the surface of the welded joints, thereby improving its fatigue life. This type of surface enhancement technology can significantly improve the surface morphology of the welded joints, reduce the geometric discontinuity of the structure, and introduce beneficial compressive stress, thereby reducing residual tensile stress [4,5].

UIT is an effective method for improving the fatigue performance of welded joints. This method has gained significant research attention in recent years. Numerous researchers have analyzed the mechanical properties, microstructure, and residual stress of welded structures to investigate the optimization effects of UIT.

In terms of mechanical properties, UIT has proven to be highly effective in enhancing the fatigue life of welded joints [6,7,8,9,10,11,12,13,14]. Studies have shown that UIT can optimize the performance of welded components composed of various materials and structures, such as high-speed steel [15] and S700 steel [16], resulting in a fatigue strength increase of over 50%. It is noted that the effectiveness of UIT is influenced by several factors, including duration, amplitude, and temperature. For instance, Yu et al. [17] conducted UIT on the surface of 16MnR steel butt joint welding parts, finding that longer impact times led to greater increases in fatigue life. The growth rates at treatment durations of 30 and 60 min reached 375.22% and 521.24%, respectively. Chen et al. [18] carried out UIT on the welded joints of 7A52 (Al-Zn-Mg-Cu) alloy and analyzed the influence of time parameters on the UIT effect.

In addition, UIT not only improves the geometric profiles of the welding seams and reduces the stress concentration factor [19], but it also optimizes the microstructure of the welded joints, and then improves the overall mechanical properties. Researchers have determined that UIT can generate grain refinement, such as forming a modified layer on the surface of the 2219 aluminum alloy friction stir welding (FSW) joint with plastic deformation morphology and small grains [20]. This process dissolves larger precipitates and improves the joint’s stress corrosion cracking resistance. Using UIT to treat the circumferential welding seam of austenitic stainless steel can form a grain refinement layer and a transition layer, and significantly reduce the number of precipitates in the transition layer. This improves the microhardness and maximum shear strength by approximately 18.9% and 9.6%, respectively [21]. Wang et al. [22] conducted a microstructural study on 7075 aluminum alloy and its friction-welded joints. After UIT, the specimens developed a gradient refinement structure known as the ultra-fine grained (UFG) layer, which is capable of enhancing the material’s hardness and strength. Currently, researchers have confirmed that UIT achieves its optimization effects through mechanisms such as grain refinement and a reduction in the number of precipitates. Further, artificial intelligence is greatly helpful in detecting changes in the strength and toughness of microstructure and chemical composition changes [23]. These findings provide new ideas for deepening our understanding of UIT’s effects.

Finally, UIT has the ability to improve the residual stress state on the surface of welded joints. Residual stress refers to the stress within a material caused by uneven deformation and alterations to its microstructure during processes such as material processing, heat treatment, and welding. These residual stresses can greatly affect the performance and fatigue life of the material. UIT eliminate the residual stress on material surfaces by applying compressive stress and thermal stress, thereby forming a uniform, stable, and resilient surface. By utilizing high-frequency energy, UIT can alter the grain dislocations of the materials, thereby increasing their density and reducing the overall energy. This process leads to the release of residual stress, providing a more stable stress state for the material surface [24,25]. For example, Hu et al. [26] successfully eliminated residual stress in the weld and heat-affected zones of 316L weld joints using UIT, enhancing the quality and strength of the welding. This also occurred when stainless steel plates were treated. Liu et al. [27] conducted UIT on a localized area of the circumferential weld of stainless steel pipes. The influence of localized UIT on the stress distribution of the weld was studied: UIT introduces compressive stresses on the impacted surface; it also affects the stress distribution within the circumferential weld of the stainless steel pipes up to a depth of 8 mm, forming a compressive stress layer with a depth of 4 mm in the treated area. This technology is not limited to butt joints; it is also suitable for fillet welding in orthotropic steel bridge decks. Yuan et al. [28] developed a 3D numerical simulation method to analyze the effect of UIT on the fatigue strength of the welded joints of orthotropic bridge decks.

This study was conducted using a combined approach of experiment and finite element analysis. The study focuses on analyzing the effect of UIT on the fatigue performance of T-joints, as well as studying the changes in the angle misalignment, microstructure grain, and surface micro-hardness of T-joints before and after undergoing UIT. Taking into account the notch effect, the influence of the transition angle change of the T-joint after UIT on its stress concentration is discussed using the traction structural stress method.

## 2. Materials and Methods

### 2.1. Preparation of Test Specimens

This study conducted experiments on double-sided welded T-joint specimens with S355J2 steel as the base material and E500T-1 as the weld wire material (Wuhan Second Navigation Engineering Bureau in China, Wuhan, China). The chemical composition and mechanical properties of the base metal and weld wire are shown in Table 1 and Table 2, respectively.

The specific preparation process of test specimens is shown in Figure 1. The T-joint specimens were prepared from orthotropic steel bridge decks, and the machining process was mainly divided into two steps: interior and exterior welding seams of U-ribs, and wire cutting.

(1)According to the size and welding process of the orthotropic steel bridge decks used in the Hong Kong-Zhuhai-Macao Bridge, the U-rib specimens of the bridge decks were fully automatically welded by the Wuhan Second Navigation Engineering Bureau. The welding sequence involved simultaneous welding of the interior welding seam, then stepwise welding of the exterior welding seam (see Figure 2). The welding parameters were current 300–320 A; voltage 30 V; the welding speed for the first pass (backing welding) was 300 mm/min, and the welding speed for the second pass (cover welding) was 350 mm/min. At the same time, during the welding process, the two ends of the U-rib top plate of the bridge panel were clamped to restrict their movement.(2)The 2000 mm long orthotropic bridge deck was divided into six U-rib fatigue test specimens of 300 mm width by wire cutting equipment; then, the U-rib fatigue test specimens with 100 mm width were obtained by continuous cutting. Finally, the T-joint specimens used in this study were obtained by local wire cutting.

The length of the abdominal plate was 70 mm, with a thickness of 8 mm. The angle between the support plate and the base plate was 78°, as shown in Figure 3.

### 2.2. Traction Structural Stress Method

#### 2.2.1. Basic Principles

The traction structural stress method involves determining the potential failure section of the welded joints based on the stress condition and calculation of the membrane stress and bending stress through static equilibrium conditions to obtain the structural stress. The structural stress of the welded structure can be accurately determined through the finite element software Abaqus 6.14 [29,30].

To calculate the structural stress using the finite element method, the welded T-joints should be modeled, and constraints and loads should be set based on the clamping condition of the test specimens in the fatigue test. As depicted in Figure 4, the nodal forces at the weld toe were extracted using the established local coordinate system, and the membrane stress, bending stress, and structural stress were computed using Formulas (1)–(3) [31,32].
(1)$$\sigma_{m}=\frac{1}{t\prime }\sum_{i=1}^{n}F_{x\prime ,i}$$
(2)$$\sigma_{b}=\frac{6}{{\left(t\prime \right)}^{2}}\sum_{i=1}^{n}F_{x\prime ,i}\left(y-\frac{t\prime }{2}\right)$$
(3)$$\sigma_{s}=\sigma_{m}+\sigma_{b}$$
where: *t*′—corresponding crack path length, *F_x_*_′,*i*_—nodal forces at nodal position *i* along the crack path, *y*—distance in the direction of cracking at the nodal point, *σ_s_*—structural stress, *σ_m_*—membrane stress, *σ_b_*—bending stress.

#### 2.2.2. Validation of Mesh Insensitivity

The traction structural stress method has mesh insensitivity. In the finite element simulation process, only the geometric features of the weld toe of the T-joints need to be fully considered, and a rougher mesh can be used to obtain a relatively accurate result.

In order to verify the mesh insensitivity of the structural stress method, three FE models with mesh sizes of 4 mm, 2 mm, and 1 mm were used to calculate the structural stress of T-joints.

The finite element modeling, meshing, constraints, and boundary conditions of the T-joints are shown in Figure 5. The 70 mm long area at the left and right ends of the models is limited Y-direction displacement, and the rotation and movement of the left end of the models in X and Z directions are constrained, ensuring that the boundary conditions in the finite element simulation process are consistent with the actual fatigue test. A load of 1 MPa was applied to the right end of the test specimens. As shown in Figure 6, the trend of the traction structure stress curve at the same stress extraction position under three different mesh sizes was basically the same, and the maximum stress of the traction structure was 1.0831 MPa, which verifies the mesh insensitivity of the traction structure stress method.

### 2.3. Ultrasonic Impact Treatment (UIT) Method

Ultrasonic impact treatment (UIT), similar to shot blasting and hammering, is a type of surface treatment method. It applies high-frequency impact heads onto the surface of T-joints, producing compression plastic deformation on the surface layer and changing the original stress field. The high-energy impact rapidly increases the surface temperature of the specimen, thereby changing the surface structure and optimizing the surface shape and residual stress of the weld zone. As shown in Figure 7, the UIT equipment generally consists of an ultrasonic generator, transducer, amplitude modulator, and ultrasonic impact head. Before the ultrasonic impact treatment, the appearance surface of the treatment area should be clean and intact to establish a path that allows the ultrasonic impact head to move smoothly.

In this study, the UIT-300 ultrasonic impact gun was used to treat the weld toe of the T-joints. The diameter of the impact needle was 4 mm, the working frequency of the ultrasonic impact gun was 17.7 kHz, and the amplitude was 25 μm. First, the specimen was fixed on the workbench. Then, the impact needle was aligned with the weld toe, making sure that the impact zone at the weld seam was consistent with the base metal. The axis of the impact gun was located on the centerline of the transition angle between the weld toe and the base metal. The impact needle repeatedly impacted along the longitudinal direction of the weld toe until a clear trace was formed, as shown in Figure 8.

### 2.4. Measurement of Welding Seam Size and Angular Misalignment

In order to further study the effect of UIT on the weld size of T-joints, laser measurement instrument Wiki Scan (SERVO-ROBOT Inc., Hocquart Saint-Bruno, QC, Canada) was used to measure the weld size of the test specimens in this study. The results of the fillet weld of the T-joint specimens are shown in Figure 9 and Table 3. It can be found that UIT produces an arc-shaped notch with a depth of 0.2 mm at the weld toe.

During the welding process, the uneven heat input caused shrinkage in the thickness direction of the T-joints, thereby leading to angular misalignment α of the joints, as shown in Figure 10. In fatigue tests, additional tensile stress and bending stress are generated on the weld toe and root of the T-joints, reducing the fatigue life of the welded joints. Therefore, when evaluating the fatigue performance of welded joints, it is necessary to consider the influence of angular misalignment α. For the measurement of the angular misalignment α of T-joints, the contour profiles of test specimens are firstly scanned in 3D; then, the contour shapes are imported into computer-aided design (CAD) and the angular misalignment α of the specimens is obtained using its built-in measuring tools. Figure 11 shows the measurement results of angular misalignment α of the specimens in this study. Compared with the original welded T-joints, the angular misalignment α of the test specimens after UIT was significantly reduced by approximately 50%. Meanwhile, there were obvious arc-shaped notches at the exterior weld toe, and the transition angle between the weld and the base material was significantly improved.

### 2.5. Metallographic Structure Observation

As shown in Figure 12, after the processing was completed, the corresponding method was utilized to prepare the sample for observing the microstructure of the processed area. The microstructure sample was L-shaped, with a length and width of 20 mm and a thickness of 10 mm. The sample was prepared perpendicular to the processing direction, and the cross section should include the area that underwent ultrasonic impact treatment. Prior to observing the microstructure, it is necessary to grind the cross section using sandpaper in the order of 400 #, 600 #, 800 #, and 1000 #, followed by polishing with a black damping cloth using SiO_2_ suspension until the scratches disappear; then, polishing with silk velvet cloth using water until the specimen changes color, and subsequently corroding it with a specific reagent. Optically, clear grain boundaries can be observed under a microscope, and metallographic photos of each zone can be taken.

### 2.6. Hardness Test

The microhardness test was conducted on the metallographic specimen. The specimen was placed on the loading table of the microhardness tester and the loading device was used to press the indenter into the shape of a pyramid. The load could be increased or decreased based on the hardness of the material being tested. Once the diamond indenter is pressed into the specimen, a pit is formed on the surface. The microscope cross wire is aligned with the pit, and the length of the pit diagonal is measured using the eyepiece micrometer. By considering the load and the length of the pit diagonal, the microhardness value of the measured material can be calculated.

## 3. Results

### 3.1. Metallographic Analysis of T-Joints after UIT

In this study, the metallographic samples were prepared, ground, and polished until the scratches disappeared; then, the samples were etched with 4% nitrate alcohol solution for about 15 s.

The metallographic samples were observed using an OLYMPUSDSX510 optical digital microscope to analyze the effect of UIT on the microstructure of the joints. Figure 13 illustrates the microstructure of the S355J2 steel-welded joint. It can be found that the welded joint of fusion welding is mainly composed of weld metal (WM), heat-affected zone (HAZ), and base metal (BM), as shown Figure 13a. HAZ consists of two distinct zones, namely, coarse grain HAZ (CGHAZ) and fine grain HAZ (FGHAZ) [33]. Figure 13b illustrates that the depth of UIT’s influence on this welded joint is approximately 0.2 mm.

In Figure 13c, the WM is mainly composed of block- and sheet-like proeutectoid ferrite. Pearlite precipitates around the strip like ferrite, which means there is a small amount of pearlite next to the proeutectoid ferrite organization. In Figure 13d, the CGHAZ has ferrite distributed in blocks. The microstructure of FGHAZ contains ferrite, pearlite, and a small amount of granular bainite, as shown Figure 13e. Figure 13f shows the microstructure of the FGHAZ affected by UIT. Compared with Figure 13e, it can be observed that the grains exhibit a compressed behavior, and ferrite, pearlite, and bainite, which are arranged more closely.

Microhardness tests were conducted on the ARTCAM-300SSI-C digital turret microhardness tester. During the microhardness test, the applied load was 300 g and the load holding time was 15 s. The distribution of the hardness testing points is shown in Figure 14, with two hardness testing paths crossing the ultrasonic impact zone and fusion line in an L shape, with a parallel spacing of 1 mm. The distance between each test point is 0.5 mm.

According to the hardness curve in Figure 14b, from the hardness curve of path 2, it can be observed that the HAZ in the lower right of the fusion line exhibits a softening behavior, and the hardness decreases by approximately 10%. Comparing the hardness curves of paths 1 and 2, it is possible to observe the influence of UIT, in which hardness increased by approximately 10%, and reaching the maximum value being around 278 HV near the fusion line.

### 3.2. Fatigue Tests

In this study, a total of nine test specimens were used for fatigue testing: three ultrasonic impact specimens (D-1, D-3, D-5) with a stress ratio of 0.1; three ultrasonic impact specimens (D-2, D-4, D-6) with a stress ratio of 0.3; and three as-welded T-joints (D75 series) with a stress ratio of 0.1 [34], as shown in Table 4. During the fatigue testing process, a 100-ton MTS Landmark dynamic fatigue testing machine was employed, and both ends of the T-joints were clamped using hydraulic wedges. The clamping width and length can be seen in Figure 15.

The failure location and the macroscopic fracture surface of the test specimens are shown in Figure 16, respectively. Multiple fatigue cracks were initiated on the surface of the interior weld toe of all specimens and propagated along the plate thickness and plate width. With the increase in fatigue cycles, the fatigue cracks continued to expand until the thickness of the T-joint was close to the thickness of the through joint. The ductile fracture occurred when the bearing area of the welded joints was insufficient.

## 4. Discussion

### 4.1. The Effect of the Notch on the Traction Structural Stress of the Weld Toe

According to the microstructure of the test specimens in Section 3.1, it can be clearly seen that the weld toe of the specimen forms an obvious arc-shaped notch after UIT. The conventional calculation method of the traction structural stress at the weld toe does not take into account the impact of the notch. Therefore, it is necessary to consider the notch effect on the traction structural stress at the weld toe of the T-joints, so as to meet the traction structural stress obtained when there is a notch on the actual weld toe.

Next, an analysis of the structural stress at the weld toe of the specimen with a plate thickness of t and a crack depth of l caused by the notch was conducted, as shown in Figure 17. The overall structure is in a self-balancing state at the cross-section A-A, and equivalent stresses $p_{m}$ and $p_{b}$ are defined on the crack surface. Assuming that the self-balancing stress caused by the gap can be calculated based on any crack depth *l* using the principle of equilibrium, the equivalent stresses represented by $p_{m}$ and $p_{b}$ can be redistributed and calculated using $\sigma_{m}^{t},\;\;\sigma_{b}^{t},\;\sigma_{m},\;\sigma_{b}$ in another cross-sectional area (at a given gap depth $t_{1}$), thus obtaining the equivalent structural stress at the crack notch. The calculation formula is as follows:
(4)$$a=\frac{t_{1}}{t}$$
(5)$$p_{m}=(\frac{1}{2a}-\frac{3}{2}+a)\sigma_{m}^{t}-(\frac{1}{2a}-\frac{3}{2}+a)\sigma_{b}^{t}-(\frac{1}{2a}-\frac{5}{2}+a)\sigma_{m}+(\frac{1}{2a}-\frac{1}{2})\sigma_{b}$$
(6)$$p_{b}=(\frac{1}{2a}+\frac{1}{2}-a)\sigma_{m}^{t}-(\frac{1}{2a}+\frac{1}{2}-a)\sigma_{b}^{t}-(\frac{1}{2a}+\frac{1}{2}-a)\sigma_{m}+(\frac{1}{2a}+\frac{1}{2})\sigma_{b}$$
(7)$$p_{s}=p_{m}+p_{b}$$
where: $\sigma_{m}^{t}$—far field membrane stress, $\sigma_{b}^{t}$—far field bending stress, $\sigma_{m}$—membrane stress at a given crack depth $t_{1}$, $\sigma_{b}$—bending stress at a given crack depth $t_{1}$, $p_{m}$—membrane stress on the crack surface, $p_{b}$—bending stress on the crack surface, $p_{s}$—structural stress on the crack surface.

The above only provides the calculation formula for $p_{m}$, $p_{b}$. However, the calculation of $\sigma_{m}$, $\sigma_{b}$ requires introducing a bilinear distribution with characteristic depth to estimate the equivalent stress of the self-balancing part (dashed line) under the stress state. Figure 18a, and Figure 18b shows the stress distribution of the self-balancing part, which can be considered as two linear distributions in regions 1 and 2, respectively. By conducting equilibrium conditions and traction continuity at point 2, the following formula can be obtained [35,36,37,38]:(8)$$\sigma_{1}=\frac{1}{2}\cdot (2\cdot \sigma_{2}^{\left(1\right)}+\sigma_{1}^{\left(1\right)})-\frac{t_{1}}{2\cdot t}\cdot \left(2\cdot \sigma_{2}^{\left(1\right)}+\sigma_{3}^{\left(2\right)}\right)$$
(9)$$\sigma_{2}=\sigma_{2}^{\left(2\right)}+\frac{t_{1}}{t}\cdot \left(\sigma_{2}^{\left(1\right)}-\sigma_{2}^{\left(2\right)}\right)$$
(10)$$\sigma_{3}=\sigma_{3}^{\left(2\right)}+\frac{t_{1}}{2\cdot t}\cdot \left(\sigma_{2}^{\left(2\right)}-\sigma_{2}^{\left(1\right)}\right)$$

In regions 1 and 2, the equivalent membrane stress and bending stress are:(11)$$\sigma_{m}=\frac{\sigma_{1}+\sigma_{2}}{2},\;\sigma_{b}=\frac{\sigma_{1}-\sigma_{2}}{2}$$

The dimension of the weld zone without UIT is shown in Figure 19a; the establishment of the finite element model and the settings of the conditions are shown as Figure 19b. Next, the traction structural stress at the interior and exterior weld toes are extracted and calculated according to the structural stress formula with the notch effect. Figure 20 portrays the relationship between the traction structural stresses at the interior and exterior welding seams and the ratio of the notch depth ($t_{1}$) to the overall specimen thickness (t). It can be observed that as the ratio $\frac{t_{1}}{t}$ increases, the structural stress at both the interior and exterior weld toes decreases, indicating the effect of the notch depth. When $\frac{t_{1}}{t}$ is less than 15%, the structural stress at the interior weld toe exceeds that of the exterior weld toe. However, when $\frac{t_{1}}{t}$ exceeds 15%, the structural stresses at both the interior and exterior weld toes become increasingly consistent and display a stable trend. When $\frac{t_{1}}{t}$ = 1, which implies that the given crack depth matches the specimen’s thickness, the structural stress at the interior and exterior weld toes reached their minimum value, equal to 1.029 MPa.

After considering the influence of the notch effect depth, it can be concluded that the structural stress variation trend of the interior and exterior weld toes is essentially consistent. It can be found that the structural stress of the exterior weld toe is slightly smaller than that of the interior weld toe.

.

### 4.2. Traction Structural Stress Calculation after the UIT

Under UIT, the weld toe morphology of the T-joint specimens changes, forming an arc-shaped notch, and the geometric transition is smoother. We overlooked the change in thickness and focused on studying the impact of the rounding radius of the face transition into the native material (transition angle). Therefore, by changing the transition angle, the effect of UIT on the structural stress at the weld toe of the joint is studied. In this section, three FE models with different weld profiles are used to study. As shown in Figure 21, the transition angles between the weld zone and the base metal are 100°, 140°, and 180°, respectively. The transition angle of 180° is similar to the morphology of the weld toe after UIT.

According to FE-safe 2019 software, the nodal stress at the weld toe of the three models was extracted, and then the structural stress was calculated according to the notch effect formula in Section 3.1. The ratio of the structural stress at the weld toe and the given notch effect depth ($t_{1}$) to the overall thickness (t) of the specimen under different transition angles $\frac{t_{1}}{t}$ was obtained, as shown in Figure 22.

As the $\frac{t_{1}}{t}$ ratio increases, the structural stress at the weld toe at different transition angles decreases and finally tends to be consistent. When $\frac{t_{1}}{t}$ is less than 15%, the structural stress at the weld toe decreases with the increase in the transition angle, and the trend is obvious. In addition, when the transition angle is 180°, the structural stress of the weld toe is the minimum, indicating that the UIT can effectively reduce the traction structural stress at the welded toe; when $\frac{t_{1}}{t}$ is 15%, the structural stress is 1.8 MPa when the transition angle is 100°, 1.6 MPa when the transition angle is 140°, and 1.35 MPa when the transition angle is 180°. If $\frac{t_{1}}{t}$ is greater than 15%, with the increase in the $\frac{t_{1}}{t}$ ratio, the traction structural stress of the weld toe under different transition angles gradually stabilizes and, finally, tends to be consistent. When $\frac{t_{1}}{t}$ =1—that is, when the given crack depth reaches the specimen thickness—the structural stress of different transition fillet toes reaches the minimum value. The structural stress is 1.029 MPa when the transition angle is 100°, 1.027 MPa when the transition angle is 140°, and 1.024 MPa when the transition angle is 180°. With the increase in the rounding radius of the face transition into the native material, the fatigue performance becomes superior due to the decrease in stress concentration factor.

### 4.3. Fatigue Performance Analysis of T-Joints after UIT

Referring to the traction structure stress curve in Section 4.1 and Section 4.2, the structural stress value of the notch effect depth/thickness ($\frac{t_{1}}{t}$) of 15% at 180° is evaluated for a chosen equivalent structural stress coefficient of 1.35.

The fatigue test results are shown in Table 5: Compared with the original welded T-joint specimens, the angular misalignment of the test specimens after the UIT is significantly reduced by about 50%; when the stress ratio is 0.1 and the nominal stress is 180 MPa, the fatigue life is up to 1,064,588. When the nominal stress is 220 MPa, the shortest fatigue life is 523,606. When the stress ratio is 0.3 and the nominal stress is 169.6 MPa, the fatigue life is up to 716,606. When the nominal stress is 220 MPa, the fatigue life is 262,582. However, for the specimens without ultrasonic impact treatment, when the stress ratio is 0.1 and the nominal stress is 169.6 MPa, the longest fatigue life is 360,021, and when the nominal stress is 233.2 MPa, the shortest fatigue life is 117,413.

Figure 23 depicts the equivalent structural stress S-N curve of the test specimens before and after UIT. The curve indicates that UIT significantly enhanced the fatigue performance of the specimen by approximately 350% according to fatigue results with a stress ratio of 0.1. The fatigue life of test specimens with a stress ratio of 0.1 is 150% higher than that of test specimens with a stress ratio of 0.3. Additionally, the fatigue life of the specimens with a stress ratio of 0.3 after UIT surpassed that of the original welded specimens with a stress ratio of 0.1, showing an increase of approximately 200%.

The curve indicates that UIT can effectively improve the fatigue life of T-joints. According to the fatigue results, UIT significantly improved the fatigue performance of the sample by about 350% at a stress ratio of 0.1. At a stress ratio of 0.3, UIT increased its fatigue life by 150%. However, as the stress ratio increases, the fatigue life of T-joint specimens gradually decreases.

## 5. Conclusions

In this study, the effect of UIT on the fatigue performance of T-joints was studied using a combination of FE simulation, theoretical analysis, and experiments. The following conclusions are drawn:(1)The metallographic analysis shows that in FGHAZ, UIT compresses the ferrite and pearlite grains, making them more tightly arranged. Through hardness testing, it was found that the HAZ in the lower right of the fusion line exhibits a softening behavior, with the hardness decreased by approximately 10%. In the area affected by UIT, there is an observed phenomenon of increased hardness, estimated to be around 10%.(2)Considering notch effect, increasing the transition angle can effectively alleviate stress concentration. When notch effect depth ($t_{1}$)/the overall thickness (t) is 15%, the structural stress at transition angles of 100°, 140°, and 180° under unit load is 1.8 MPa, 1.6 MPa, and 1.35 MPa, respectively.(3)UIT significantly increased the fatigue life of the T-joint by approximately 350% at a stress ratio of 0.1 and approximately 150% at a stress ratio of 0.3, owing to the stress concentration reduction caused by UIT.

## Figures and Tables

**Figure 1 materials-16-06196-f001:**
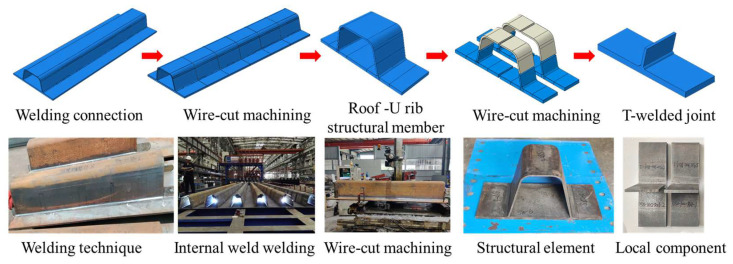
Preparation process for the fatigue test specimen of the bridge deck.

**Figure 2 materials-16-06196-f002:**
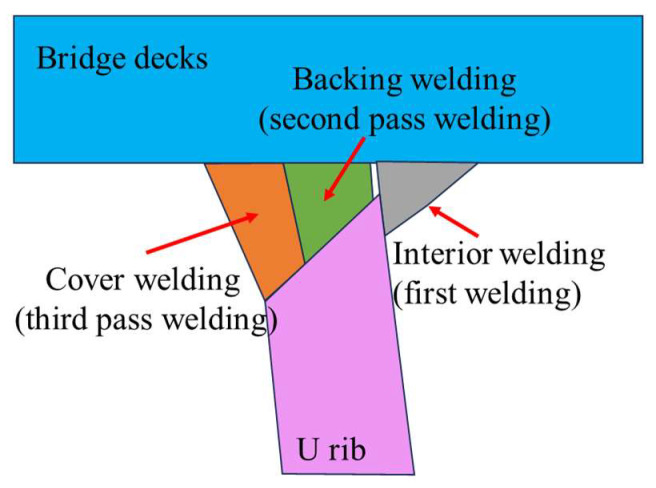
Double-sided welding process.

**Figure 3 materials-16-06196-f003:**
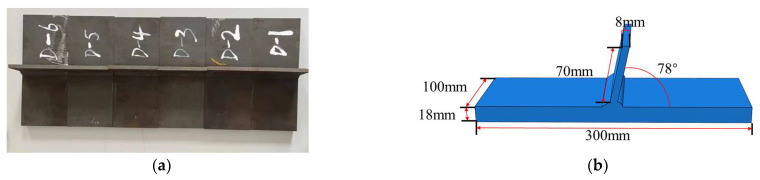
The test specimens: (**a**) T-joint specimens; (**b**) specific dimensions of the specimen.

**Figure 4 materials-16-06196-f004:**
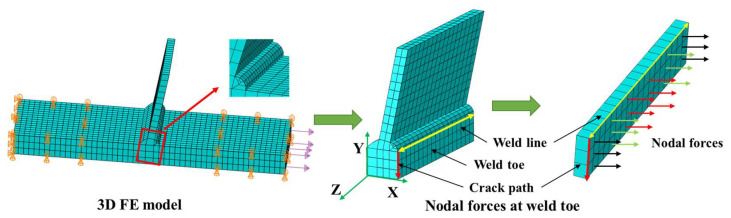
Schematic diagram of boundary condition of finite element model (FE model) and nodal forces at weld toe.

**Figure 5 materials-16-06196-f005:**
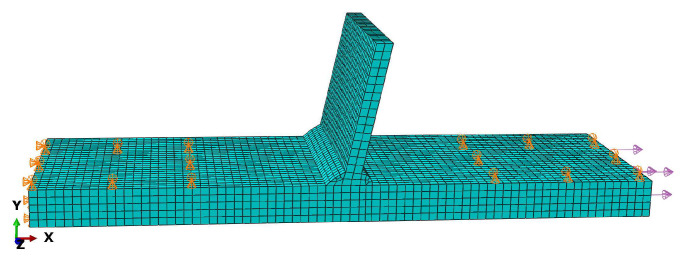
Finite element modeling, meshing, and boundary conditions.

**Figure 6 materials-16-06196-f006:**
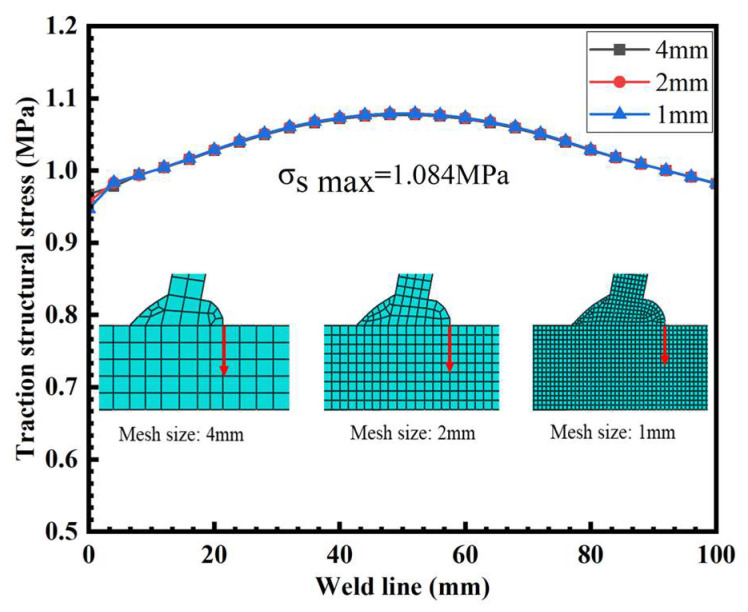
The traction structural stress of T-joints with different mesh sizes.

**Figure 7 materials-16-06196-f007:**
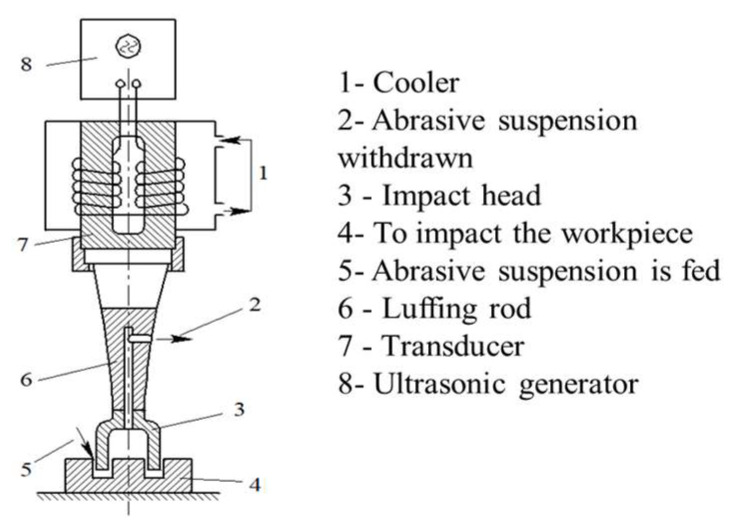
The basic structure of the UIT equipment.

**Figure 8 materials-16-06196-f008:**
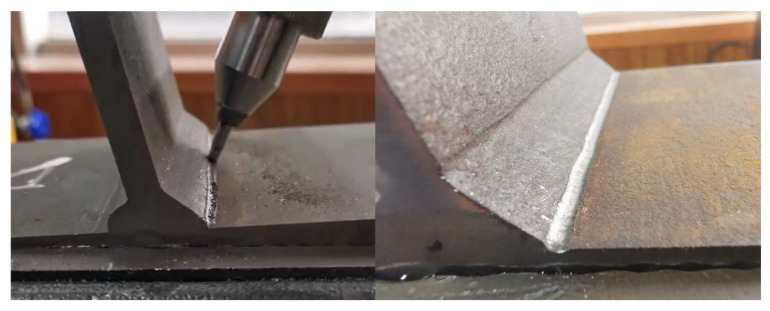
The UIT of exterior weld toe.

**Figure 9 materials-16-06196-f009:**
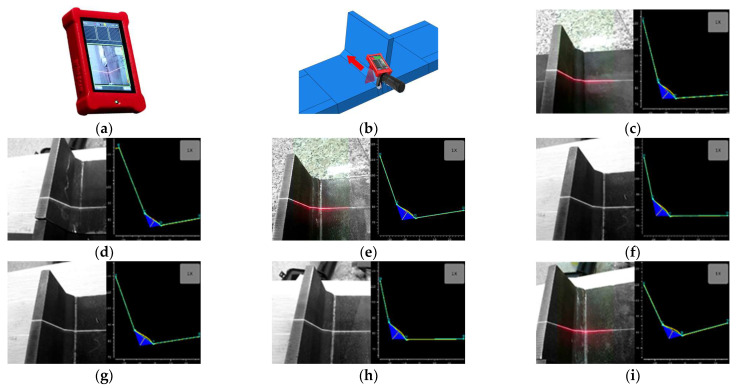
Wiki Scan scanning results; (**a**) Wiki Scan equipment; (**b**) measure position; (**c**) D75-100-R01-1; (**d**) D-1; (**e**) D-2; (**f**) D-3; (**g**) D-4; (**h**) D-5; (**i**) D-6.

**Figure 10 materials-16-06196-f010:**
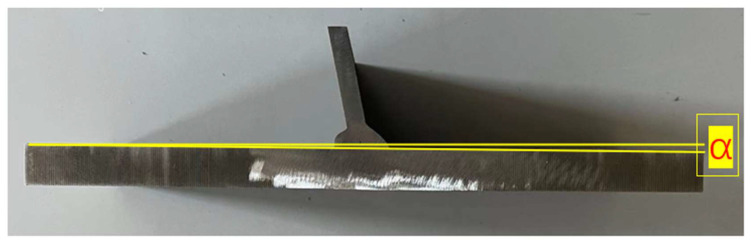
Schematic diagram of angle misalignment measurement of T-joint.

**Figure 11 materials-16-06196-f011:**
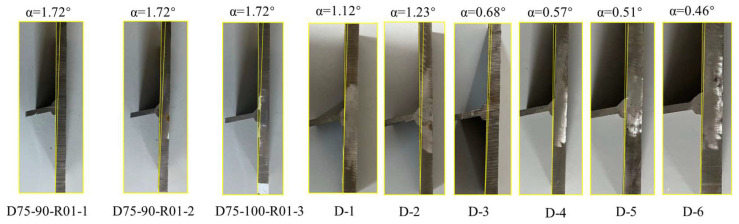
Measurement results of angle misalignment for test specimens.

**Figure 12 materials-16-06196-f012:**
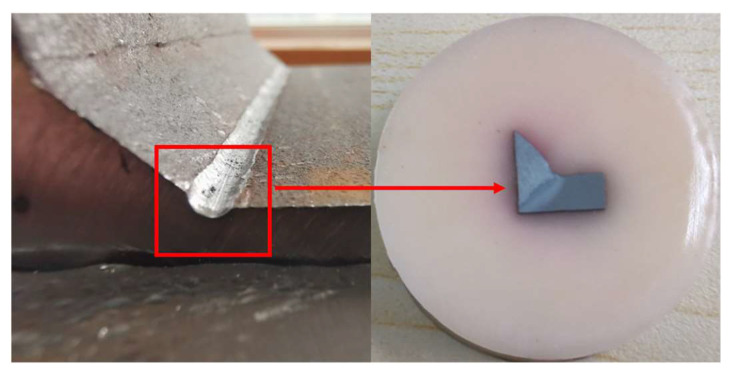
Metallographic observation sample.

**Figure 13 materials-16-06196-f013:**
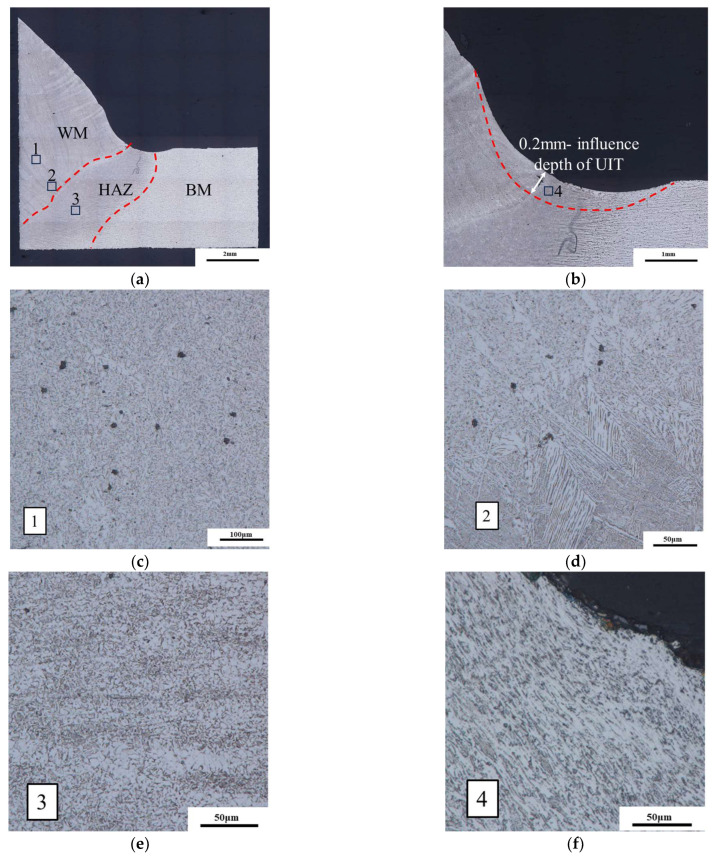
Microstructures across the T-joints. T-joint after UIT: (**a**) macrostructure of weld joint; (**b**) influence depth of UIT; (**c**) WM; (**d**) CGHAZ; (**e**) FGHAZ; (**f**) FGHAZ after UIT.

**Figure 14 materials-16-06196-f014:**
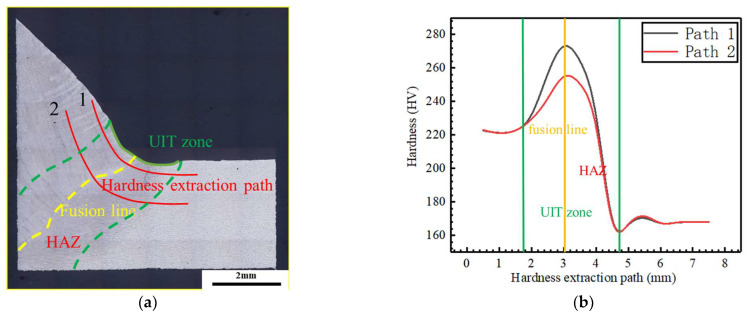
Hardness test: (**a**) Image observation of the sample’s hardness; (**b**) hardness curve of the sample.

**Figure 15 materials-16-06196-f015:**
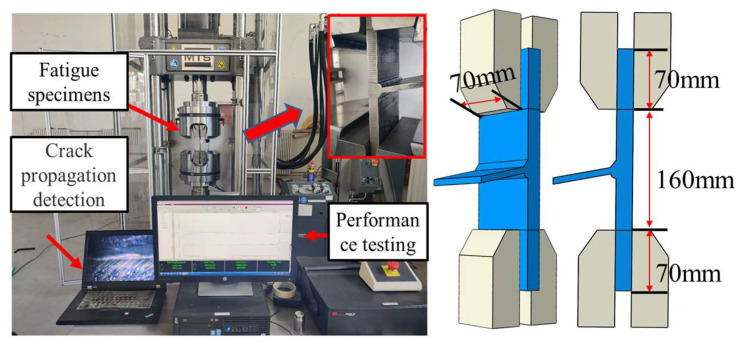
Schematic diagram of fatigue tests for test specimens.

**Figure 16 materials-16-06196-f016:**
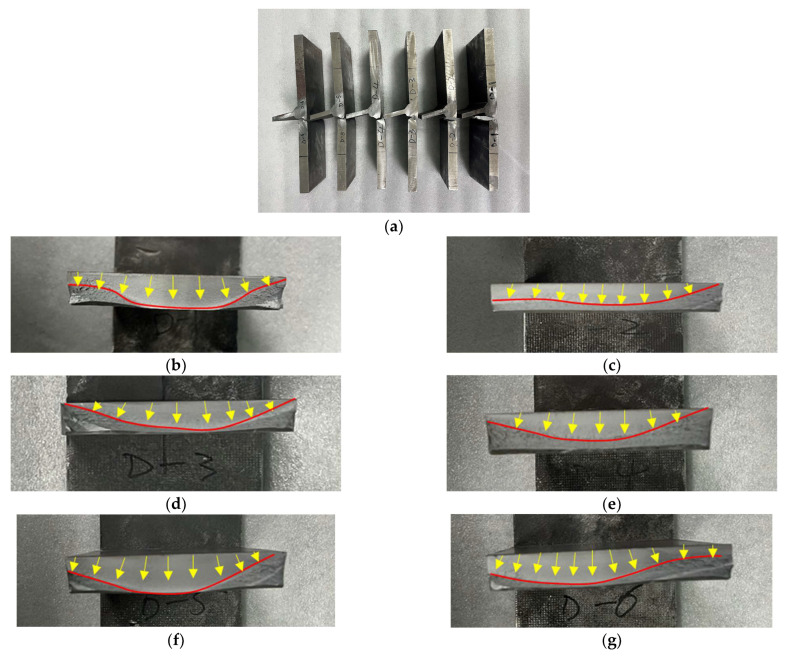
Macro-fracture characteristics of welded T-joint specimens: (**a**) the T- joints after the failure; (**b**) D-1 test specimen; (**c**) D-2 test specimen; (**d**) D-3 test specimen; (**e**) D-4 test specimen; (**f**) D-5 test specimen; (**g**) D-6 test specimen.

**Figure 17 materials-16-06196-f017:**
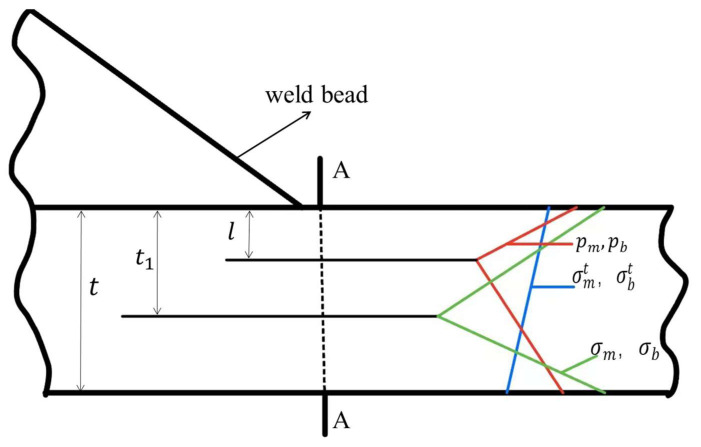
An estimation scheme for equivalent crack face pressure for a hypothetical crack at an arbitrary depth (*l*) from the notch root.

**Figure 18 materials-16-06196-f018:**
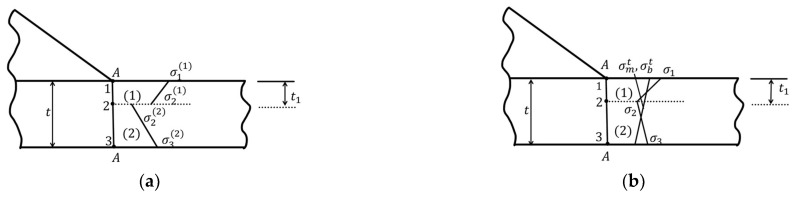
Estimation of self-equilibrating part of the stress state induced by notch: (**a**) the stress distribution before the self-equilibrating; (**b**) imposing equilibrium requirements for regions (1) and (2) and continuity at location 2 (1–3 represent three nodes, (1) is the region between node 1 and node 2, (2) is the region between node 2 and node 3).

**Figure 19 materials-16-06196-f019:**
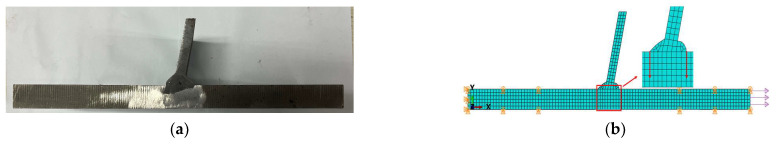
The macroscopic and FE model of T-joints: (**a**) macroscopic T-joint specimen; (**b**) finite element model, meshing and boundary condition.

**Figure 20 materials-16-06196-f020:**
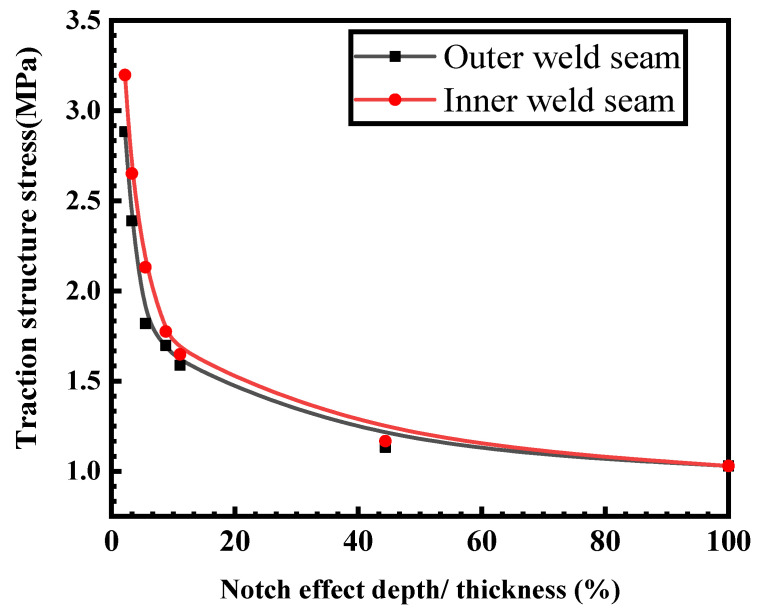
Variation of traction structure stress of interior and exterior weld with $\frac{t_{1}}{t}$.

**Figure 21 materials-16-06196-f021:**
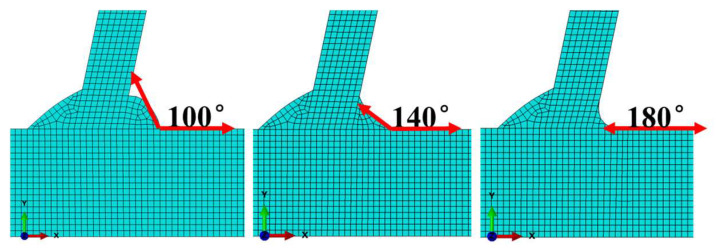
FE models of T-joints with different weld profiles.

**Figure 22 materials-16-06196-f022:**
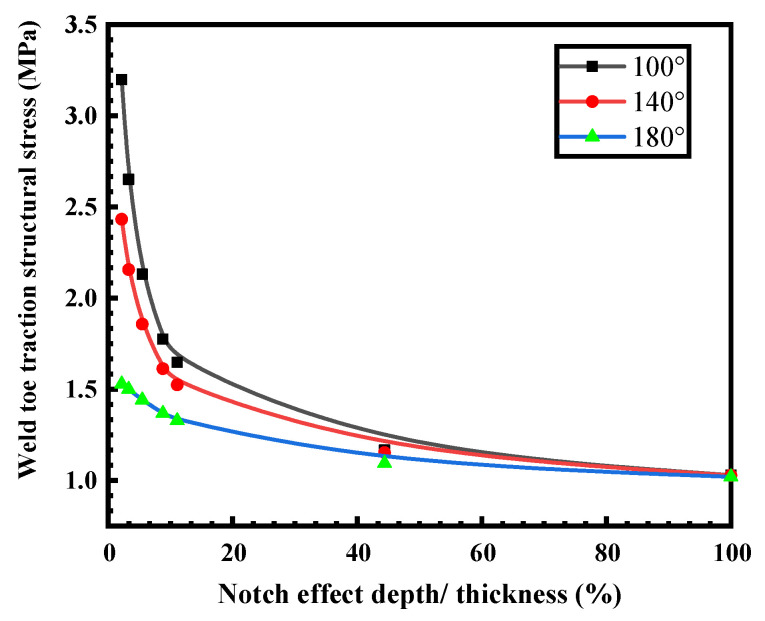
Traction structural stress corresponding to different transition angles at the external welding toe.

**Figure 23 materials-16-06196-f023:**
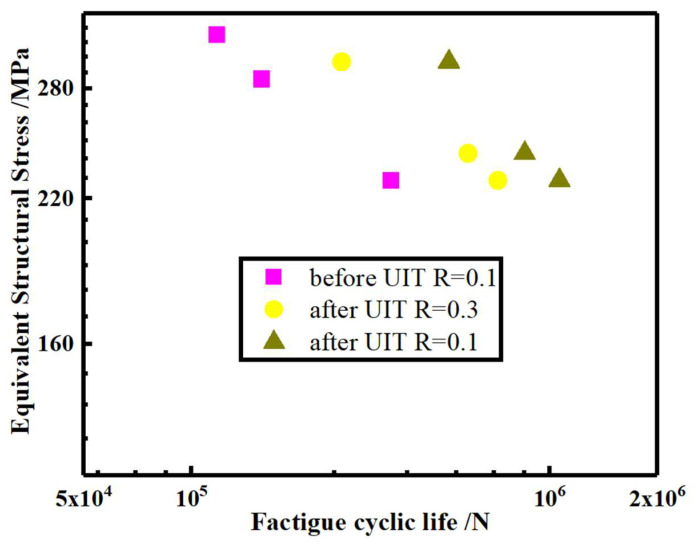
The S-N curve of the T-joint specimens.

**Table 1 materials-16-06196-t001:** Chemical composition of base metal and welding wire (wt.%).

Materials	C	Mn	Si	P	Ni	Cr	Cu
S355J2	0.083	1.32	0.37	0.011	0.17	0.48	0.31
E500T-1	0.09	1.42	0.44	0.015	0.32	0.63	0.24

**Table 2 materials-16-06196-t002:** Material properties of base metal and welding wire.

Materials	Elastic Modulus/GPa	Yield Strength/MPa	Poisson’s Ratio
S355J2	207	390	0.3
E500T-1	210	496	0.3

**Table 3 materials-16-06196-t003:** The measurement results of weld seam size and angle misalignment.

Test Specimens	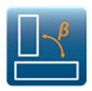	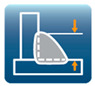	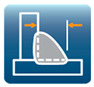	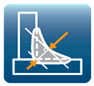	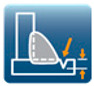
	Included Angle/°	Weld Leg1/mm	Weld Leg2/mm	Weld Throat/mm	Notch Depth/mm
D-6	104	10.5	8.6	5.8	0.2
D-5	104	9.2	7.2	4.9	0.2
D-4	104	10.5	8.9	5.8	0.2
D-3	104	11.5	9.2	6.3	0.2
D-2	104	9.8	7.5	5.2	0.2
D-1	105	9.3	7	4.9	0.1
D75-100-R01-1	104	10.1	8.4	5.9	0

**Table 4 materials-16-06196-t004:** Fatigue test parameters and results of test specimens.

Serial Number	Stress Ratio R	Nominal Stress/MPa	Frequency/Hz	Fatigue Life N/Cycles
D75-1	0.1	233.2	10	117,413
D75-2	0.1	212	10	156,790
D75-3	0.1	169.6	10	360,021
D-1	0.1	180	10	1,064,588
D-2	0.3	180	10	591,766
D-3	0.1	220	10	523,606
D-4	0.3	220	10	262,582
D-5	0.1	169.6	10	852,125
D-6	0.3	169.6	10	716,606

**Table 5 materials-16-06196-t005:** Fatigue test results of T-joints.

Test Specimen	Angular Misalignment α/°	Equivalent Structural Stress/MPa	Stress Ratio R	Fatigue Life N/Cycles
D75-90-R01-1	1.72	315	0.1	117,413
D75-90-R01-2	1.72	286	0.1	156,790
D75-100-R01-3	1.72	229	0.1	360,021
D-1	1.12	229	0.1	1,064,588
D-2	1.23	243	0.3	591,766
D-3	0.68	297	0.1	523,606
D-4	0.57	297	0.3	262,582
D-5	0.51	243	0.1	852,125
D-6	0.46	229	0.3	716,606

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
