# Peer review of "Fatigue Performance Analysis of Welded T-Joints in Orthotropic Steel Bridge Decks with Ultrasonic Impact Treatment"

_materials, 2023, doi:10.3390/ma16186196_

Round 1

Reviewer 1 Report

1.       Lines 48-49 and 58-59 and 66-67. I agree that additional treatments improve the fatigue resistance of joints, but one must also remember about the irreversible phase transformations taking place in high-strength steels, and especially the problem of the softening zone in HAZ. Such issues are indicated, for example, in the following articles (they do not have to be referenced):

https://doi.org/10.1016/j.engfailanal.2021.105502

https://doi.org/10.1515/adms-2017-0039

Moreover both shows the methodology, and the results of metallographic examination.

2.       The fineness of the weld grain is not the final solution here and there are many additional factors in addition to UIT.

3.       What's more, microhardness is not a representative quality parameter - hardness measurement (usually HV10 or HV30) is required in the assessment of unalloyed steel joints. The issue of hardness measurements of austenitic steels and aluminum alloys is a very broad issue and when evaluating the results one should always take into account the condition of the material in terms of manufacturing processes, in particular plastic working and heat treatment.

I believe points 1 to 3 above should be commented on or at least mentioned.

Fig. 2 - probably mm in spite of cm

Looking at Fig. 7, it seems to me that the welding is done in a single pass from the outside, and not from the inside and outside as predicted by the FEM model. What's more, looking at the construction of the joint, it seems to me that the sheet metal of the plate is thinner, and the welded one is thicker. Please verify this and, if necessary, replace the photo with the correct one.

How to perform UIT machining on the inside? Is there enough space and sufficient range (availability) during production to allow such processing to be performed?

Table 3 - units are missing.

There is a lack of research methodology for quality evaluation of welds and hardness measurements. To be supplemented. It is worth to consider to prepare the part where all methods will be described. Not between test results. At what load the tests were performed - information should be supplemented with the values and in Fig. 13.

Fig. 11. The presented sample for metallographic tests is either not taken from the tested joint, or the welding process was carried out incorrectly, which is indicated by the shape of the weld. Please post the correct photos.

In fig 11 or additional, please indicate the range of impact and structural changes introduced in the material. Moreover, please indicate to what depth the changes are observed.

Line 240-257 – microstructural analysis must be improved:

- What is “crystal structure of the welding wire”? Observation are carried out for weld metal, so the wire metal was molten, and solidify as weld.

- How the elongate shapes of grain reached? It is effect of what? Probably plastic deformation of weld surface.

- “fine columnar microstructure” – it is not fine, the size is an effect of crystalline grow direction and the cutting plane of specimen.

- The microstructure of weld metal shows that the crystallization is an effect only thermal conductivity from steel and cooling speed.

- Microstructure interpretation is incorrect: the observed structure is ferrite (with different shapes) and pearlite. How did you identify bainite?

The developed model has some shortcomings. This lacks a proper macroscopic photo that would show that there are sheets and a weld in the joints, and most importantly, the weld is between the sheets. What's more, why wasn't a fine mesh used in the stress concentration area? The second important piece of information that is missing is the diagram of preparing the edges for welding and the diagram of the welding sequence for the entire module (see: https://doi.org/10.1016/j.mfglet.2022.12.001). Without such a scheme, the assumed models are unrealistic to obtain - please try to justify technologically how to obtain the angle of 180°. Tests have shown that there is a notch of 0.2 mm. In my opinion, the key here is the rounding radius of the face transition into the native material, not the angle (it is worth checking how the angle is measured in welding practice).

Fig. 21 and line 355-359. I don't understand what the % values mean. At what shape of notch and what depth?

Acc. to test results and analysis – what number of cycles is required for welded joins? Where are the results of your test and analysis in comparison with the requirements?

The conclusions are very general - they should be related to the results obtained, taking into account the above comments. Pay particular attention to the dispersion of the results.

Author Response

Authors’ Item-by-Item Response to Referees’ Comments, please see attached.

Reviewer 2 Report

The presented manuscript seems to be interesting for readers of the Materials journal, it is written in a good manner and suits the requirements of the journal. It can be accepted for publication after minor corrections listed below.

- The "Abstract" section should contain the main achievements of research not general discussion. Re-organization of abstract is needed.

- State of the art to be improved further.

- What was the basis for choosing the parameters? Why are the experimental design methods not used in the selection of variables and their levels?

- Yu [17] should be changed to Yu et al. [17]

- The number of figures in the text has gone from 21 to 24. There are no figures 22 and 23. Please check and correct.

- More than a third of the references are for before 2013.

- Figure 11(a) has no special significance and can be omitted. Figure 11(b) can be merged with Figure 12.

- The subsections and caption of Figure 12 should be revised

- The dimensions in Figure 2(b) and schematic Figure 14 should be converted from centimeters to millimeters

- The specifications of D75-90-R01 and D75-100-R01 samples should be explained in the text and table.

- All parameters used in formulas must be explained. It is recommended to attach all parameters and abbreviations used in a table at the end of the article.

- Abbreviation/ acronyms, should all be defined at their first occurrence in the manuscript,

- In the "Conclusion" section, the authors should present more quantitative data as the main results of the research study rather than just some qualitative data.

- Literature review is not sufficient and authors must review and cite more papers in the field of “Changes in strength and toughness based on changes in structure and chemical composition in the weld and heat-affected zone, as well as predicting properties based on artificial intelligence“. Doing this, review and citing the following refs could be helpful: Neural Network World, 23,2, 2013, 117., Neural Network World, 23, 2013, 351-367.

The presented manuscript seems to be interesting for readers of the Materials journal, it is written in a good manner and suits the requirements of the journal. It can be accepted for publication after minor corrections

Author Response

(The authors gave the same response as above.)

Round 2

Reviewer 1 Report

Thank you very much for your answers and changes made. It seems to me that the presented material will be more transparent for the reader in its present form.

Please go though the spelling mistakes in the text.

Reviewer 2 Report

The article mighgghtt be accepted for publication.

Fine